# Evaluation of UV Aerosol Retrievals from an Ozone Lidar

Shi Kuang[1]*, Bo Wang[1], Michael J. Newchurch[1], Kevin Knupp[1], Paula Tucker[1], Edwin W. Eloranta[2], Joseph P. Garcia[2], Ilya Razenkov[2], John T. Sullivan[3], Timothy A. Berkoff[4], Guillaume Gronoff[4,5], Liqiao Lei[4,6], Christoph J. Senff[7,8], Andrew O. Langford[7], Thierry Leblanc[9], Vijay Natraj[10]

[1]University of Alabama in Huntsville, Huntsville, Alabama, USA

[2]University of Wisconsin-Madison, Madison, Wisconsin, USA

[3]NASA Goddard Space Flight Center, Greenbelt, Maryland, USA

[4]NASA Langley Research Center, Hampton, Virginia, USA

[5]Science Systems and Applications Inc., Lanham, Maryland, USA

[6]Universities Space Research Association, Columbia, Maryland, USA

[7]NOAA Earth System Research Laboratory, Boulder, Colorado, USA

[8]Cooperative Institute for Research in Environmental Sciences, University of Colorado, Boulder, Colorado, USA

[9]Jet Propulsion Laboratory, California Institute of Technology, Wrightwood, CA, USA

[10]Jet Propulsion Laboratory, California Institute of Technology, Pasadena, California, USA

*Correspondence to: Shi Kuang (kuang@nsstc.uah.edu)

## Abstract

Aerosol retrieval using ozone lidars in the ultraviolet spectral region is challenging but necessary for correcting aerosol interference in ozone retrieval and for studying the ozone-aerosol correlations. This study describes the aerosol retrieval algorithm for a tropospheric ozone lidar, quantifies the retrieval error budget, and intercompares the aerosol retrieval products at 299 nm with those at 532 nm from a high spectral resolution lidar (HSRL) and with those at 340 nm from an Aerosol Robotic Network radiometer. After the cloud-contaminated data is filtered out, the aerosol backscatter or extinction coefficients at a 30-m and 10-min resolution retrieved by the ozone lidar are highly correlated with the HSRL products, with a coefficient of 0.95 suggesting that the ozone lidar can reliably measure aerosol structures with high spatio-temporal resolution when the signal-to-noise ratio is sufficient. The actual uncertainties of the aerosol retrieval from the ozone lidar generally agree with our theoretical analysis. The backscatter color ratio (backscatter-related exponent of wavelength dependence) linking the coincident data measured by the two instruments at 299 and 532 nm is 1.34±0.11 while the Ångström (extinction-related) exponent is 1.49±0.16 for a mixture of urban and fire smoke aerosols within the troposphere above Huntsville, AL, USA.

## 1. Introduction

A tropospheric ozone differential absorption lidar (DIAL) makes measurements of vertical ozone profiles, typically at two wavelengths chosen between 277 and 300 nm with a separation less than 12 nm, by weighing several parameters such as the ozone absorption cross sections, solar background, dynamic range of the detection system, and interference from aerosols and other species (e.g., Alvarez et al., 2011; Browell et al., 1985; De Young et al., 2017; Fukuchi et al.,

001; Kempfer et al., 1994; McDermid et al., 2002; Proffitt and Langford, 1997; Strawbridge et al., 2018; Sullivan et al., 2014). Vertical aerosol profiles are of high interest not only because they are needed for aerosol correction in ozone lidar retrievals (Steinbrecht and Carswell, 1994), but also because simultaneous ozone and aerosol vertical profile measurements provide unique information on their interactions and on sources of pollutant transport (Browell et al., 1994; Langford et al., 2020; Newell et al., 1999). However, there is currently no consensus on the reliability of the aerosol retrievals produced by ozone lidars due to the difficulty of solving the three-component lidar equation and the large variability in aerosol optical properties associated with the multiplicity of aerosol types and size distributions.

The most widely used solution for the elastic single-wavelength aerosol lidar equation is the analytic method developed by Klett (1981). The inversion method then inspired Fernald (1984) to publish a computer algorithm scheme to solve the more general two-component (aerosol and molecular) atmospheric lidar equation. The Klett (1981) inversion requires *a priori* for the lidar ratio (i.e., aerosol extinction-to-backscatter ratio, represented by "$S$" hereafter) to link the aerosol backscatter with its extinction for solving the lidar equation. Lasers used for aerosol lidars are preferred in the visible and infrared bands, typically 532 and 1064 nm for Nd:YAG laser or 694 nm for Ruby laser (Russell et al., 1979), where the ozone absorption is much smaller than molecular and Mie scattering. In the ultraviolet (UV) band for an ozone lidar, the ozone absorption may not be trivial. Some ozone lidars have an aerosol channel available, either independently or sharing receiving optics with the ozone channel (e.g., Browell et al., 1994; De Young et al., 2017; Gronoff et al., 2019; Kovalev and McElroy, 1994; Uchino and Tabata, 1991). For most of the traditional two-wavelength ozone lidars without an aerosol channel, although there has been some discussion about the aerosol retrieval algorithm (e.g., Eisele and Trickl, 2005; Langford et al., 2019; Papayannis et al., 1999; Sullivan et al., 2014), the evaluation of the aerosol retrieval product and its error budget has rarely been addressed. Due to a significant wavelength difference with aerosol lidars, several aspects of the aerosol retrieval using an ozone lidar are worth noting. First, the signal-to-noise ratio (SNR) for ozone lidars decays quicker with altitude due to more significant UV molecular (i.e., Rayleigh) scattering and ozone absorption resulting in a lower retrievable altitude than aerosol lidars. Second, since the molecular and ozone components become more important at UV wavelengths compared to visible and infrared wavelengths, the uncertainties in aerosol retrieval propagated from the calculation of these two components are expected to be larger for ozone lidars than for aerosol lidars. Third, $S$ and the wavelength dependence used for the ozone lidar wavelengths may be different from those used for the longer aerosol lidar wavelengths (Ackermann 1998; Eck et al., 1999).

The primary objectives of this article are to investigate the performance of our aerosol retrieval algorithm and to quantify its error budget for the ozone lidar. The secondary goal is to seek the overall wavelength dependence between the aerosol optical properties measured by the ozone lidar at 299 nm and by a high spectral resolution lidar (HSRL) at 532 nm.

## 2. Instruments and Data Processing

### 2.1. Ozone Lidar

The Rocket-city Ozone ($O_3$) Quality Evaluation in the Troposphere ($RO_3QET$) lidar is located on the campus of the University of Alabama in Huntsville (UAH) at 34.725 ˚N and 86.645 ˚W at 206 m asl and is one of the six systems of the Tropospheric Ozone Lidar Network (TOLNet) (http://www-air.larc.nasa.gov/missions/TOLNet). This system

measures ozone from 0.1 km up to about 12 km during nighttime and up to about 6 km during daytime with a temporal
resolution of 2 min. The vertical resolution of the lidar retrievals varies from 150 m in the lower troposphere to 750
m in the upper troposphere in order to keep the measurement uncertainty within ±10% (Kuang et al., 2013).
The transmitter comprises two Raman-shifted lasers at 289 and 299 nm. Two 30-Hz, 266-nm Nd:YAG lasers
pump two 1.8-m Raman cells, respectively, with mixtures of active gas and buffer gas to generate 289 and 299-nm
lasers with an average pulse energy of about 5 mJ. The receiving system consists of three receivers with diameters of
2.5 cm, 10 cm, and 40 cm, respectively, and four photomultipliers (PMTs) similar to that described by Kuang et al.
(2013) except that the solar filters have been replaced by 300-nm short-pass filters for all telescopes. Channels-1, 2,
3, 4 represent the 2.5-cm, 10% of the 10-cm, 90% of the 10-cm, and the 40-cm telescope channels, respectively. Since
the modification of Channel–4 through the addition of narrow-band solar filters was not completed before the time
period of this study, data from this channel was not used in this work, with the net result that uncertainties for ozone
retrievals above 6 km during daytime were often too large due to the strong solar background. Lidar signal counting
was accomplished by four Licel transient recorders (Licel company, Germany) with both analog and photoncounting
(PC) modes, with a sampling rate of 40 MHz corresponding to a 3.75-m fundamental resolution. The cloud base height
is determined by the following empirical method. Derivatives of the logarithm of the off-line analog signal are
calculated for a lidar signal profile and the first range bin at which the derivative is greater than a certain threshold is
considered to be the cloud base height. The threshold is chosen empirically based on the lidar SNR and the vertical
resolution. Therefore, lidar data with cloud base lower than 2 km was discarded. The cloud filtering process should
be conducted carefully because an elastic lidar without a polarization channel is not capable of accurately
distinguishing aerosols and clouds solely through their backscatter properties. Five 2-min lidar data intervals were
combined to give a 10-min lidar-signal integration time to improve the SNR. Further, six of the 3.75-m fundamental
bins were integrated for all channels. In addition, dead-time correction (for PC signal only), background correction,
analog-PC signal merging, and signal-induced noise correction were performed.
**2.2. Aerosol Retrieval and Uncertainty Estimation**
The aerosol profiles were retrieved with an iterative DIAL algorithm. A brief description of this algorithm is provided
in this section, with further details in Appendix A. A first-order Savitzky-Golay differentiation filter with a second-
degree polynomial was applied to the logarithm of the signal ratios to compute the first-cut ozone profile. This initial
ozone profile was substituted back into the three-component lidar equation to derive the profile of aerosol backscatter
coefficients at 299 nm by assuming a constant $S$ of 60 sr and boundary value of the aerosol backscatter coefficient at
a far-range reference altitude, about 10 km. During the daytime, the ozone retrieval was limited by the lower SNR of
the 289-nm channel, but the 299-nm channel had much better SNR due to lower atmospheric extinction and was able
to measure aerosol up to higher altitudes. $S$ has large variability as a function of aerosol characteristics, humidity, and
wavelength (Ackermann, 1998; Strawbridge et al., 2018; Mishchenko et al., 1997). The $S$ *a priori* value assumed for
this study represents a mix of urban and smoke aerosols during the lidar observations (Ackermann, 1998; Burton et
al., 2012; Cattrall et al., 2005; Groß et al., 2013; Müller et al., 2007). The *a priori* is application dependent. In the
aerosol retrieval uncertainty discussion in Appendix B, we assume a ±20% uncertainty for $S$ based on an average
standard deviation obtained from prior observations (Müller et al., 2007).
Molecular backscatter and extinction profiles were computed from local radiosonde data. Then, the aerosol
profile was substituted into the lidar equation again to obtain a stable solution, usually within three iterations. This
aerosol profile was further employed to calculate the aerosol correction for ozone retrievals using the first-order Taylor
approximation (Browell et al., 1985) by assuming a power-law wavelength dependence for the aerosol extinction and
choosing an appropriate Ångström exponent. Since this work focuses only on aerosol retrieval, details of the ozone
correction will be described in a future article. Finally, the aerosol profiles derived by the three altitude channels were
merged into a single profile in the overlapping altitude zones, i.e., 0.5–1 km for Channels-1 and 2 and 1.5–2 km for
Channels-2 and 3.
The primary uncertainty sources for the aerosol lidar retrievals are the uncertainties in lidar signal
measurement, boundary value assumption for aerosol backscatter coefficient, air density measurement, $S$ a priori, and
ozone profile input. The relative importance of these sources is altitude dependent. In the planetary boundary layer
(PBL) where the air is typically turbid, the $S$ uncertainty is dominant while other sources are minor (only few percent).
The uncertainty of $S$ influences the uncertainty of the aerosol backscatter through a complicated relationship. However,
the magnitude of the above two uncertainties can be approximately seen to be close. At the far range (higher than 7
km), lidar signal detection noise and inaccurate boundary value assumption are important. Influence from both of the
above sources, especially the boundary value, on the aerosol retrieval quickly decreases towards the ground from the
far range. In the middle range (PBL top – 7 km), both the air density measurement error and lidar signal detection
noise are essential. Uncertainty due to ozone profile input is relatively unimportant and is only few percent at most
altitudes. Figure B1 presents an example of the aerosol backscatter uncertainty calculated from 10-min nighttime
RO$_3$QET lidar data. The error budget estimate generally justifies the choice of using 6 km as the maximum altitude
for RO$_3$QET-HSRL comparison since the total uncertainty for the RO$_3$QET aerosol retrieval could be unacceptably
large (i.e., persistently larger than 100%).
**2.3.     HSRL**
The University of Wisconsin HSRL (Eloranta, 2005) was deployed in Huntsville, AL from 19 June to 4 November
2013 and operated almost 24 hours every day to support the Studies of Emissions and Atmospheric Composition,
Clouds and Climate Coupling by Regional Surveys SEAC[4]RS campaign (Kuang et al., 2017). The HSRL transmitter
was a diode-pumped Nd:YAG laser at 532 nm with a pulse energy of about 50 μJ and a pulse repetition frequency of
4 kHz. The expanded laser beam was transmitted coaxially with a 40-cm telescope with a tiny field of view (FOV) of
100 μrad to reduce solar background. The HSRL spectral filtering can separate the molecular backscatter from the
aerosol backscatter due to the molecular Doppler broadening effect, while the particulate backscatter remains
spectrally unbroadened. Aerosol backscatter coefficients can then be calculated as the difference between the total
return and the molecular component (Grund and Eloranta, 1991). In principle, aerosol extinction can be computed by
comparing the measured attenuated molecular backscatter to a reference, unattenuated molecular backscatter profile
that is calculated from the radiosonde-measured air density profile or a numerical model (Hair et al., 2008). However,
small and fast signal fluctuations were found in the partial overlap region (between the surface and about 4.5 km) for
the data taken in Huntsville so that aerosol extinction below 4.5 km cannot be derived with satisfying precision. The
signal fluctuations were probably caused by small optical misalignments from temperature changes within the lidar
system (Reid et al., 2017). The aerosol backscatter calculation is not affected by the lidar signal fluctuations since any
range-dependent instrument effects are canceled out. Therefore, we focus on the aerosol backscatter intercomparison
between the HSRL and RO$_3$QET. If aerosol extinction is needed for the HSRL, we will calculate it from the aerosol
backscatter by assuming a constant lidar ratio. The HSRL provides aerosol products with a 30-m vertical resolution
and 1-min temporal resolution from near the surface to 15 km. To achieve sufficient SNR for both HSRL and ozone
lidar and to reduce the uncertainty arising from the clock bias of the controlling computers, we adopt 10-min temporal
average and 30-m vertical average for both HSRL and ozone lidar in the intercomparison study. The HSRL has a
backscatter measurement precision better than $10^{-7}$ (m sr)$^{-1}$ for a 1-min signal average (Reid et al., 2017), which
represents an estimated precision for the extinction coefficient of better than $2\times10^{-6}$ m$^{-1}$ for a 10-min average.
**3.   Intercomparison Results**
We select four time periods (21–23 June, 14–15 August, 27–28 August, and 5–6 September 2013) to investigate the
ozone lidar capability for measuring aerosol column and range-resolved profiles. All four cases have coincident ozone
lidar and HSRL observation periods longer than 24 hours, fully covering the convective mixing layer development
and collapse processes (Klein et al., 2019) and having significant smoke layers in the free troposphere. Due to the
significant extinction and potential multiple scattering caused by clouds, the ozone lidar is incapable of measuring
either ozone or aerosol accurately above clouds, especially thick clouds. Therefore, data contaminated by clouds is
filtered out. At this time, the narrow-band interference filters had not been incorporated into the receiving system, and
the wide-band filter resulted in substantial solar background during the daytime; hence, we set 6 km asl as the
maximum altitude for intercomparison. The uncertainty of the aerosol retrieval owing to lidar signal measurement
error is dominant at far range and is determined by the lidar SNR, as shown in Appendix B.2. The solar background
is an important noise resulting in the lidar signal measurement error during daytime and is partly responsible for the
high aerosol retrieval uncertainty above 6 km as shown by the example in Figure B1. The 10-min HSRL profiles are
interpolated to the times of the ozone lidar data.

First, we investigate the correlation of the integrated (or column) aerosol backscatter between the ozone lidar

and HSRL to obtain a general relationship between their averages. Figure 1 shows that the RO$_3$QET- and HSRL-
derived integrated backscatter coefficients for all four cases are highly correlated, with a Pearson correlation
coefficient of 0.99. The root mean square error (RMSE), the standard deviation of the residuals, is negligibly small at
$1\times10^{-3}$ sr$^{-1}$, suggesting that the linear regression equation can accurately represent the relationship between the AOD
measured by the two instruments. The 493 sampling profiles cover 82 hours of coincident ozone lidar and HSRL
observations. We define the aerosol backscatter color ratio ($\mathring{a}_\beta$) as (Burton et al., 2012):
$$\mathring{a}_\beta = -\frac{d(ln\beta_A)}{d(ln\lambda)} = -\frac{\ln(\frac{\beta_A^{299}}{\beta_A^{532}})}{\ln(\frac{299}{532})}, \tag{1}$$

where $\beta_A^{299}$ and $\beta_A^{532}$ represent the aerosol backscatter coefficient at 299 and 532 nm, respectively. The subscript "$A$"
represents the "aerosol" component, to be distinguished from the "molecular" contribution that is represented by
subscript "$M$" in the Appendix. $\mathring{a}_\beta$ is an exponent denoting backscatter-related wavelength dependence, to be
distinguished from the commonly-used Ångström exponent (Ångström, 1929) that refers to the wavelength
dependence of optical thickness or extinction coefficient. $\mathring{a}_\beta$ is also different from another often-used concept, "color
ratio of the lidar ratios", which refers to the ratio of $S$ at two different wavelengths. The slope of the regression, equal
to 2.16, results in the best least-squares fit value of 1.34 for $\text{å}_\beta$ at 299 and 532 nm. The uncertainty of the column $\beta_A^{299}$
is expected to be smaller than the uncertainty for $\beta_A^{299}$ at a particular altitude and for a 10-min integration time (in
Figure B1) since the average over longer time and altitude range greatly reduces the random noise as suggested by the
small RMSE in Figure 1. If the uncertainty of the column $\beta_A^{299}$ measurements is estimated to be 20% which is
primarily due to the uncertainty of the $S$ *a priori* (a systematic error), we can estimate the corresponding uncertainty
for $\text{å}_\beta$=1.34 to be ±0.11 by error propagation from Eq. (1). $\text{å}_\beta$ has important applications in aerosol type classification
from (spectral) aerosol lidar measurements (e.g., Cattrall et al., 2005; Hair et al., 2008; Müller et al., 2007). There is
significant variation in $\text{å}_\beta$ for 532–1064 nm reported in different studies, with numbers ranging from negative values
to 2.3 (Burton et al., 2012; Cattrall et al., 2005; Müller et al., 2007). However, all of these studies show a larger value
of $\text{å}_\beta$ for smoke and urban aerosols than for maritime and dust aerosols. Since most previous studies report $\text{å}_\beta$ for
wavelengths longer than 355 nm, $\text{å}_\beta$ calculated in this study for 299–532 nm could provide valuable data for UV
wavelengths.
In practice, aerosol extinction is a more meaningful parameter and more relevant for several applications than
backscatter. For the HSRL, the extinction coefficients are linearly converted from the backscatter coefficients by
assuming a constant $S = 55$ sr with 20% uncertainty, in the same manner as Reid et al. (2017). The estimated Ångström
exponent for 299 and 532 nm is 1.49±0.16, using the data in Figure 1 after considering uncertainties in $S$ for both
lidars. The calculated Ångström exponent is different from the backscatter-related wavelength exponent because of
the wavelength dependence of $S$. The Ångström exponent from this study (1.49±0.16) is within a reasonable range
compared to previous studies. For example, the Ångström exponent was measured by a Raman lidar to be between
1.35±0.2 and 1.56±0.2 at 355 nm for smoke aerosols in Canada (Strawbridge et al., 2018). The Ångström exponent
for urban aerosols was measured to be 1.4±0.5 in Europe and 1.7±0.5 in North America for 355 and 532 nm (Müller
et al., 2007).
The Aerosol Robotic Network (AERONET) (Holben et al., 1998) provides aerosol optical depth (AOD)
measurements in eight spectral bands between 340 and 1020 nm with a temporal resolution of about 15 min. The
measurement uncertainty for AERONET AOD is within 0.02 and is expected to be larger in the UV bands (Eck et al.,
1999; Holben et al., 2001). Even though the measurement is at a different wavelength, the AERONET AOD at 340
nm can provide an additional constraint for the choice of $S$ for the RO₃QET aerosol retrieval, especially since both
instruments are at the same location. Figure 2 presents the intercomparison of the RO₃QET lidar derived AOD at 299
nm and all available AOD data at 340 nm (Smirnov et al., 2000) from the collocated AERONET sun-sky radiometer
(data for 21–23 June is unavailable). The near-surface region is assumed to be homogeneous and assigned the same
aerosol extinction values as the lowest available 30-m layer from the RO₃QET retrievals. Then, the aerosol extinction
coefficients are integrated from 0 to 6 km asl to calculate the lidar-derived AOD. The omission of aerosol extinction
above 6 km and the homogeneity assumption for the near-surface region are sources of bias for the comparison since
the AERONET instrument measures the total column AOD. The lidar has more data and higher temporal resolution,
therefore, the lidar-derived AOD is interpolated to the AERONET measurement times. Figure 2 shows that the AOD
retrieved by the two instruments has a correlation coefficient of 0.97 and a small RMSE for a total duration of about
31 hours. The mean percentage difference between the RO$_3$QET and AERONET AOD is 15±9%. The *S a priori*
directly affects the AOD calculation. The lidar-derived AOD is on average 15% larger than the AERONET AOD due
to the shorter wavelength of the lidar measurement, suggesting that the choice of $S = 60$ sr is appropriate. For a rough
estimation, the 1-σ standard deviation (9%) of the differences can be considered as the uncertainty of $S$ if the variability
of these differences are mostly due to the variation in $S$. Considering that AERONET measures the column average
AOD, with longer temporal integration, has its own uncertainty, and covers only 38% of the total observational period,
our assumption for $S = 60±20$% sr is appropriate for RO$_3$QET lidar profiling measurements with higher temporal and
vertical resolution and should be good enough to cover various uncertainty sources. The collocated AERONET data
enhances the credibility of our lidar aerosol retrieval and help evaluate the *S a priori*, with the caveat that the 124
paired data covering 31 hours is not a large sample. We do not show HSRL-AERONET comparison here since Reid
et al. (2017) have done so using more extensive data in a visible band taken at the UAH site in summer 2013.
Figure 3 presents the intercomparison of the aerosol backscatter retrieved by the HSRL and the RO$_3$QET
lidar for the four cases in 2013. The HSRL-derived aerosol backscatter coefficients are scaled to 299 nm (represented
by "HSRL-converted" hereafter) using the best-fit exponent value å$_\beta$=1.34. Some clouds lower than 2 km show up in
the HSRL curtains but not in the RO$_3$QET curtains (e. g., 1500–2100 on 15 August and 1500–2100 on 28 August).
These low-cloud-contaminated data were discarded in the RO$_3$QET lidar pre-processing program since the ozone lidar
probes the atmosphere with a shorter wavelength than the HSRL, and is, therefore, more affected by cloud interference.
Profiles with clouds higher than 2 km measured by the RO$_3$QET were retained, and the aerosol retrievals below the
clouds were used for the range-resolving intercomparisons.
In terms of the aerosol measurement evaluation, we pay attention to the two capabilities of the RO$_3$QET lidar:
measuring the PBL diurnal evolution and measuring free-tropospheric smoke layers. In Figure 3, the PBL heights
measured by the two lidars, which are identified by large aerosol gradients, are highly consistent for all cases. The
development of the convective mixing layer in the early morning, an important process responsible for surface ozone
increase, can be visually identified in most RO$_3$QET curtains (e.g., 1400–1700 UTC or 0900–1200 local time in Figure
3-h). The aerosol structures and evolution in the free troposphere measured by the RO$_3$QET lidar are highly similar to
those measured by the HSRL. For example, the RO$_3$QET lidar captured an extremely thin aerosol layer at ~5-km
altitude on 27–28 August (Figure 3-g), which probably originated from the Pacific Northwest fire and has been
discussed by Reid et al. (2017). The large aerosol uncertainties for the RO$_3$QET lidar at far ranges are consistent with
expectation. As demonstrated in Appendix B, aerosol retrieval uncertainties due to lidar signal measurement error and
the boundary value chosen at the reference altitude, two of the most important sources of uncertainty, increase with
altitude and may exceed 100% at ~7 km.
To evaluate the range-resolving capability of the ozone lidar for aerosol retrieval, we intercompared the
aerosol backscatter coefficients, for all cases, from the two instruments with a 10-min temporal resolution and a 30-
m vertical resolution after filtering out cloud-contaminated data, as shown in Figure 4. The high correlation coefficient
of 0.95 suggests that the RO$_3$QET lidar can capture the aerosol variability with high spatio-temporal resolution. The
correlation coefficient (0.95) between the two high vertical resolution retrievals is slightly lower than that between the
RO$_3$QET and column-averaged HSRL retrievals (0.99, see Figure 1) due to less vertical average. The HSRL-converted
backscatter is calculated using $å_\beta=1.34$ and the regression equation in Figure 1. We expect the slope of the data in
Figure 4 to be very close to 1. However, the actual slope is 1.08, reflecting the fact that there are a large fraction of
points with small aerosol backscatter and larger residuals in clean air (low aerosol) regions. This is not surprising since
the HSRL has higher measurement precision than the $RO_3QET$ lidar so that their relative differences in clean air
regions can be large.

Figure 5 presents the mean and 1-σ standard deviations of the relative differences between $RO_3QET$ and

HSRL retrievals, ($RO_3QET$-HSRL)/HSRL, to be compared with the theoretical 1-σ error calculated as outlined in
Appendix B. The HSRL measurements are considered as the "true" values to be compared with the $RO_3QET$
measurements. Both the theoretical and actual 1-σ values generally increase with altitude. The actual differences
between $RO_3QET$ and HSRL measurements are mostly within or of comparable order of magnitude to the theoretical
calculation of the $RO_3QET$ measurement uncertainties. The structures of the theoretical uncertainties are consistent
with the actual differences at most altitudes, with few exceptions. For example, the large discrepancies (red lines
compared to blue lines in Figure 5) occurring at ~4.5 km in Figure 5 (c) and ~1.5 km in Figure 5 (d) are primarily
because of small number division effects for the extremely clean atmospheric layers (also see Figure 3). Aerosol
backscatter of clean air can be accurately measured by the HSRL, but may be beyond the measurement sensitivity of
$RO_3QET$.

In Figure 5, the $RO_3QET$-measured aerosols are generally higher than the HSRL-measured aerosols between

5 and 6 km so that the $RO_3QET$-HSRL differences are biased to positive altitude values. These positive biases can be
caused by two reasons. First, the $RO_3QET$ derived aerosol extinction above 5 km is obviously larger than that from
HSRL during daytime due to the solar background impact, which is especially strong in the summer. The relative
differences are even worse in clean (compared to turbid) regions during the daytime because of the small number
division effect mentioned earlier. It can be seen from Figure 3 that $RO_3QET$ nighttime retrievals above 5 km and
daytime retrievals below 5 km are relatively good due to either lower solar background or larger lidar signal resulting
in better SNR. There were both clean and smoky layers between 5 and 6 km for the four cases; therefore, the positive
differences cannot be explained solely by the lower capability of $RO_3QET$ for measuring clean air. We hypothesize
that another reason causing the positive differences between 5 and 6 km is the underestimated backscatter color ratio
for the smoke aerosols. We converted the HSRL backscatter from 532 to 299 nm using a constant backscatter color
ratio, 1.34, which represents an average for the column-integrated backscatter. The most significant contribution to
integrated backscatter comes from PBL aerosols, which are mostly urban aerosols with a lower backscatter color ratio
than either fresh or aged smoke (Burton et al., 2012; Cattrall et al., 2005). The uncertainty of the backscatter color
ratio was not considered in the error budget of the aerosol retrieval. In addition, we ignored the measurement
uncertainty of the HSRL. Therefore, the general agreement of theoretical estimates of aerosol retrieval uncertainties
and the actual errors suggests that our analysis of the uncertainty sources in Appendix B is reasonable.
**4. Conclusions**
We have evaluated the aerosol retrievals at 299 nm from the $RO_3QET$ ozone lidar using both aerosol retrievals at 532
nm from the University of Wisconsin HSRL and AERONET AOD data at 340 nm from coincident observations at
Huntsville, AL in 2013. The integrated backscatter coefficients below 6 km asl from $RO_3QET$ and HSRL are highly
correlated, with a Pearson coefficient of 0.99 after excluding cloud-contaminated data. The aerosol profiles of
backscatter coefficients at 30-m vertical and 10-min temporal resolution retrieved by RO₃QET are also highly
correlated with those from the HSRL with a coefficient of 0.95 suggesting that the ozone lidar is capable of providing
reliable aerosol structure information at high spatio-temporal resolution. Intercomparison of the backscatter product
was performed to avoid additional uncertainty caused by the lidar ratio ($S$) assumption needed for the HSRL aerosol
extinction retrieval. The RO₃QET-measured AOD below 6 km asl is also highly correlated with the AERONET-
measured AOD, with a correlation coefficient of 0.97. The 340-nm band of the AERONET AOD data is closest to the
ozone lidar wavelength among the available instruments and can, therefore, provide a constraint for the $S$ assumption
for the ozone lidar. Analysis of the intercomparison of AERONET and RO₃QET data confirms that our choice of $S =$
60 sr at 299 nm is appropriate.      The aerosol retrieval algorithm and its error budget are shown in the Appendix.
The primary uncertainty sources for the aerosol lidar retrieval are errors in lidar signal measurement, boundary value
assumption, air density calculation, $S$ *a priori*, and ozone profile input. The uncertainty in $S$ assumption is a dominant
source at near range, while the lidar signal measurement and boundary value errors dominate at far range, as shown
in Figure B1 for a sample scenario. Within the middle range (PBL top – about 7 km), the air density calculation error
is essential and is larger or comparable to the lidar signal measurement error. The total uncertainty generally increases
with altitude from about 15% in the PBL to consistently higher than 100% above 7 km. Theoretical estimates of the
error budget are generally consistent with RO₃QET-HSRL measurement differences.
By assuming a constant $S$ of 60 sr at 299 nm, the backscatter coefficients measured by RO₃QET and HSRL
are related by a backscatter color ratio (backscatter-related exponent) of 1.34±0.11 for 299 and 532 nm. The extinction-
related Ångström exponent, which is more relevant for various applications, is estimated to be 1.49±0.16 by assuming
$S = 55$ sr for the HSRL at 532 nm. These exponents represent a summertime average for a mixture of urban pollution
and fire smoke. Separation of aerosol types was not done in this work, although we recognize that $S$ and Ångström
exponent vary with the aerosol phase function and size distribution. Aerosol correction for ozone lidar retrievals will
be described in a subsequent paper.

**Appendix A. Aerosol retrieval algorithm**
The ozone DIAL solution can be written as follows:

$$n_{(r)} = \frac{-1}{2\Delta\sigma} \times \frac{d}{dr}\left[ ln\frac{P_{\text{on}(r)}}{P_{\text{off}(r)}}\right] + [B] + [E],$$           (A1)

where $n_{(r)}$ is the ozone number density at range $r$; $\Delta\sigma$ is the differential ozone absorption cross section; $P_{\text{on}(r)}$ and $P_{\text{off}(r)}$
are the backscattered on-line and off-line lidar returns; and $[B]$ and $[E]$ represent the differential backscatter and
extinction terms (Browell et al., 1985), respectively, including both molecular and aerosol components. The first term
of the right side of Eq. (A1) is often called the signal term. The subscripts "on" and "off" represent 289 and 299 nm
in this study. The aerosol extinction coefficients at 299 nm are calculated using the following procedure.
1)      A first-order Savitzky-Golay differentiation filter with a second-degree polynomial and variable fitting
window widths are applied on $ln\frac{P_{\text{on}(r)}}{P_{\text{off}(r)}}$ to compute the signal term. This smoothing method can accommodate the
rapid decay of the lidar signal with altitude to provide sufficient SNR for ozone retrievals by appropriate selection of
smoothing window widths (Leblanc et al., 2016a).
2)        By canceling the lidar constant using the two lidar equations at range $r$ and $r+\Delta r$ for 299 nm, the aerosol
backscatter coefficients at range $r$ can be expressed as (Uchino et al., 1980):
$$\beta_A(r) = -\beta_M(r) + \frac{Z(r)}{Z(r+\Delta r)}[\beta_A(r+\Delta r) + \beta_M(r+\Delta r)]\exp\left\{-2\Delta r[\alpha_A\left(r+\frac{\Delta r}{2}\right) + \alpha_M\left(r+\frac{\Delta r}{2}\right) + \alpha_{O3}\left(r+\frac{\Delta r}{2}\right)]\right\} \quad ,$$
(A2)
where $\beta_A(r)$ and $\beta_M(r)$ are aerosol and molecular backscatter coefficients at range $r$, respectively; $Z(r) = P_{off}r^2$ is
the range-corrected lidar signal at 299 nm; $\alpha_A(r+\Delta r/2)$, $\alpha_M(r+\Delta r/2)$, and $\alpha_{O3}(r+\Delta r/2)$ represent the average aerosol,
molecular, and ozone extinction coefficients between range $r$ and $r+\Delta r$, respectively. Assuming the 299-nm lidar ratio,
$S=\alpha_A/\beta_A$, is constant with the range at 60 sr for this study and further assuming:
$$\alpha_A(r + \frac{\Delta r}{2}) \approx \alpha_A(r + \Delta r) = S\beta_A(r + \Delta r),$$    (A3)
Eq. (A2) contains only two unknown variables: the aerosol backscatter coefficient $\beta_A(r+\Delta r)$ and ozone extinction
coefficient $\alpha_{O3}(r+\Delta r/2)$, which requires knowledge of the ozone number density $n_{(r+\Delta r/2)}$. Molecular backscatter and
extinction can be computed from nearby radiosonde data or a model with acceptable accuracy. For the first iteration
step, $n_{(r+\Delta r/2)}$ can be computed from the signal term in Eq. (A1). By assuming a start value $\beta_A(\text{ref})$ at a reference
range and a constant $S$ with range, $\beta_A(r)$ can be solved by Equation (A2). Then, the first $\beta_A(r)$ profile is substituted
back into (A2) to compute the second estimate by using a more accurate form for $\alpha_A(r+\Delta r/2)$ as:
$$\alpha_A\left(r + \frac{\Delta r}{2}\right) = S[\beta_A(r + \Delta r) + \beta'_A(r)]/2,$$    (A4)
where $\beta'_A(r)$ represents the value from the first estimate. Typically, a stable solution for $\beta_A(r)$, which does not change
significantly from one iteration step to the next, can be obtained with only three iterations of Eq. (A2) and (A4).
3)        The correction terms, [$B$] and [$E$], in Eq. (A1) are calculated by the Browell et al. (1985) approximation,
assuming a power-law dependence with wavelength for the aerosol extinction and choosing an appropriate Ångström
exponent. Since this paper focuses only on aerosol retrievals, the details of the ozone corrections will be described in
a future article.
4)        Aerosol profiles computed for the three altitude channels are finally merged to a single profile in their
overlapping altitude zones, 0.5–1 km for Channels-1 and 2, 1.5–2 km for Channels-2 and 3.

**Appendix B. Error budget of the aerosol retrieval**
Now we investigate five primary error sources affecting each term on the right side of Eq. (A2). In the following
section, we use the notation $\Delta$ to represent the absolute uncertainty and $\delta$ to represent the relative uncertainty. For a
function $Y$, derived from several measurement variables $x_1$, $x_2$, …, the uncertainty in $Y$ can be estimated by the
following expression using the first-order Taylor expansion approximation when these variables are independent
(Taylor, 1997):
$$\Delta Y^2 = (\Delta x_1 \frac{\partial Y}{\partial x_1})^2 + (\Delta x_2 \frac{\partial Y}{\partial x_2})^2 + \cdots .$$    (B1)
**B.1 Lidar signal measurement error**
The error source to determine the normalized lidar signal ratio term $\frac{Z(r)}{Z(r+\Delta r)}$ is the lidar signal measurement error, $\Delta P$.
Although $\Delta P$ may be due to various processes such as inaccurate dead-time correction, inaccurate background
subtraction, and signal-induced noise, its dominant component is the lidar signal statistical uncertainty (often called
lidar signal detection noise) and is typically assumed to obey a Poisson distribution. Assuming no error in deciding $r$,
by using Eq. (A2) and (B1) we obtain the uncertainty of the aerosol backscatter owing to lidar signal measurement
error, $\Delta \beta^{sig}_A(r)$, relative to the total backscatter as:

$$\frac{\Delta \beta^{sig}_A(r)}{\beta_A(r)+\beta_M(r)} = \sqrt{[\delta P(r)]^2 + [\delta P(r + \Delta r)]^2}, \tag{B2}$$

where $P(r)$ represents lidar signal counts at $r$ after omitting the wavelength subscript (i.e., 299 nm) and $\delta P(r)$ is just
the inverse of SNR. Eq. (B2) means that the uncertainty of the aerosol backscatter coefficient due to lidar signal
measurement is determined by the lidar SNR similarly to other remote sensing detection techniques. Consequently,
its relative uncertainty can be written as:

$$\delta \beta^{sig}_A(r) = \left(\frac{1}{B(r)} + 1\right)\sqrt{[\delta P(r)]^2 + [\delta P(r + \Delta r)]^2}, \tag{B3}$$

where $B(r)= \beta_A(r)/\beta_M(r)$ is the aerosol-to-molecular backscatter ratio. As expected, $\delta \beta^{sig}_A(r)$ has a reverse relationship
with $\beta_A(r)$ since it is a relative uncertainty. Figure B1 shows an example of the uncertainty budget for a 10-min lidar
data profile. The aerosol retrieval uncertainty due to the lidar signal measurement error generally increases with
altitude primarily because of the rapidly decaying lidar SNR.
**B.2 Boundary value error**
According to Eq. (A2), the uncertainty of the aerosol backscatter at $r$, $\beta_A(r)$ , can be induced by the uncertainty of the
backscatter at $r + \Delta r$, $\beta_A(r + \Delta r)$, due to the iterative computation method. The error propagation between the
adjacent altitudes can be determined by their partial differential relationship. Using the traditional far-end solution by
assuming that the air is clean at a reference altitude, the aerosol uncertainty due to the inaccurate boundary value
assumption propagates downward based on the following equation:

$$\delta \beta^{BV}_A(r) = \delta \beta_A(r + \Delta r)\left[\frac{1+\frac{1}{B(r)}}{1+\frac{1}{B(r+\Delta r)}}\right]\{1 - 2S\Delta r \beta_A(r + \Delta r)[1 + \frac{1}{B(r+\Delta r)}]\}. \tag{B4}$$

The yellow line in Figure B1 represents the relative uncertainty of backscatter retrieval due to the boundary value
assumption, $\delta \beta^{BV}_A(r)$, when $\delta \beta_A(r_b) = 1000\%$ (i.e., 10 times overestimate at $r_b$ =10 km). Despite a large
overestimate at the reference altitude, $\delta \beta^{BV}_A(r)$ decreases toward the ground, to less than 10% below 5.5 km and less
than 1% below 3.5 km. Simulations demonstrate that $\delta \beta^{BV}_A(r)$ for an underestimation of $\delta \beta_A(r_b)$ (not shown) is better
than that for an overestimation, indicating that the boundary value is preferred at a smaller value. As suggested by Eq.
(B4), $\delta \beta^{BV}_A(r)$ is affected by both $S$ and $B$. Larger $S$ (if it is correct) results in smaller $\delta \beta^{BV}_A(r)$ and, therefore, aerosol
retrieval errors converge to zero faster. In other words, the smaller the value of $S$ is, the more sensitive the aerosol
retrieval is to the boundary value error. $\delta \beta^{BV}_A(r)$ decreases with an increase of $B(r)$. This means that $\delta \beta^{BV}_A(r)$ is less
affected by the assumed value of $\beta_A(r_b)$ when the aerosol backscatter becomes more important relative to molecular
backscatter, which occurs at longer wavelengths or under turbid air conditions. It is to be noted that $\delta \beta_A(r_b)$ is
between -1 and $+\infty$ so that the distribution of $\delta \beta^{BV}_A(r)$ is asymmetric with the zero axis.
In terms of the influence of the boundary value error, we have compared our calculation with an analytical
solution proposed by Kovalev and Moosmüller (1994) (not shown); the results are almost identical. Aerosol retrieval
uncertainty due to incorrect boundary value assumption tends to converge to zero towards the lidar. It is negligible at
lower altitudes, especially in the PBL, when the air is turbid.
**B.3 Air density error**
According to Eq. (A2), the air density profile affects $\beta_M(r)$, $\beta_M(r+\Delta r)$, and the optical depth (or transmittance).
Similarly, we can derive the relative uncertainty in aerosol backscatter owing to the uncertainty in the air density
profile as:
$\delta\beta^{AD}_A(r)$
$$= \sqrt{\left\{\frac{\delta\beta_M(r)}{B(r)}\left[1 + S_m\Delta r\beta_A(r) + S_m\Delta r\beta_M(r)\right]\right\}^2 + \left\{\frac{\delta\beta_M(r+\Delta r)\left[\frac{1}{B(r)}+1\right]}{B(r+\Delta r)+1}\left[1 - S_m\Delta r\beta_A(r+\Delta r) - S_m\Delta r\beta_M(r+\Delta r)\right]\right\}^2}$$
. (B5)
$S_m$ represents the molecular extinction-to-backscatter ratio, which is a constant ($8\pi/3$). The two parts in the square root
are the components due to the uncertainties at $r$ and $r+\Delta r$, respectively. Each component includes the influences from
both molecular backscatter and optical depth. When $\Delta r$ is small, the contribution of the optical depth error is much
smaller than that of the molecular backscatter error so that (B4) can be approximated as:
$$\delta\beta^{AD}_A(r) \approx \sqrt{2}\frac{\delta\beta_M(r)}{B(r)}. \tag{B6}$$
It is to be noted that $\Delta\beta_M(r)$ and $\Delta\beta_M(r+\Delta r)$ are independent errors as assumed in Eq. (B1). If they are correlated, Eq.
(B5) will partly cancel out with their covariance term, which is not shown in (B1). Due to the nature of the iterative
computation method, $\delta\beta^{AD}_A(r+\Delta r)$ affects $\delta\beta^{AD}_A(r)$ as noted in Eq. (B4), so that the aerosol retrieval uncertainty due
to air density error will propagate downward. However, model simulation suggests that the systematic error of the air
density calculation has little impact on the aerosol retrieval because of the cancelation of the effect at $r$ and $r + \Delta r$. Eq.
(B6) means the uncertainty in the calculation of molecular backscatter will mostly linearly propagate to aerosol
backscatter. If the 2-$\sigma$ precision of a radiosonde is 0.3 K and 0.5 hPa for temperature and pressure measurements
(Hurst et al., 2011), the propagated uncertainty onto molecular backscatter is only about 0.1%. However, the real
disturbance of an atmosphere deviating from the actual air density profile may be more significant since there are
usually only a few radiosonde profiles available every day. Hence, we assume $\delta\beta_M(r)$ to be 1%, and the resulting
aerosol retrieval uncertainty is represented by the green line in Figure B1. $\delta\beta^{AD}_A(r)$ can be tens of percent in the free
troposphere and is an important error source for aerosol retrievals (Russell et al., 1979). $\delta\beta^{AD}_A(r)$ is less than 10% in
the PBL because of more turbid air in that region. Since $\delta\beta_M(r)$ is assumed to be a constant in this example, the
variation of $\delta\beta^{AD}_A(r)$ is mostly a result of varying $B(r)$, the aerosol-to-molecular backscatter ratio. Since $B(r)$ generally
increases with an increase in wavelength, $\delta\beta^{AD}_A(r)$ is expected to be smaller at longer wavelengths. Therefore, the
aerosol retrieval is less sensitive to the air density error at longer wavelengths.
**B.4 Lidar ratio error**
By using Eqn. (A2) and (B1), the relative uncertainty in aerosol backscatter due to incorrect lidar ratio ($S$) assumption
can be calculated as follows:
$$\delta\beta_A^S(r) = 2\left[\frac{1}{B(r)} + 1\right]\Delta S\beta_A(r)\Delta r. \tag{B7}$$

$\delta\beta_A^S(r)$ due to $\Delta S$ at only range $r$ appears to be small, about 1%, when $\Delta r$ is specified at 22.5 m. However, $\Delta S$ varying
with altitude is mostly systematic and, therefore, $\delta\beta_A^S(r)$ at every altitude will propagate downward, and these effects
will accumulate. The error accumulation is not straightforward to compute as an analytical solution. However, these
effects can be simulated numerically. $S$ is highly variable, and it is difficult to estimate its actual uncertainty range. In
this study, we assume that $\delta S = 20\%$ (or $\Delta S = 12$ sr) according to both previous study (Müller et al., 2007) and the
analysis using the collocated AERONET AOD data at 340 nm. The light-blue line in Figure B1 shows that the
accumulative uncertainties in the aerosol backscatter due to $\Delta S$ using Eq. (B7) and (B4) are close to the assumed 20%
uncertainty for $\delta S$. $\delta\beta_A^S(r)$ is the largest error source in the PBL which is the near range of the lidar measurement.
$\delta\beta_A^S(r)$ decreases with an increase in wavelength because of increasing $B(r)$. In other words, $\delta\beta_A^S(r)$ is less sensitive
to $\Delta S$ at longer wavelengths.
**B.5 Ozone error**
Similar to $S$, the ozone uncertainty affects only the transmittance term in Eqn. (A2) and its error propagation on aerosol
backscatter retrieval can be expressed as:
$$\delta\beta_A^{O3}(r) = 2\left[\frac{1}{B(r)} + 1\right]\Delta\alpha_{O3}(r)\Delta r. \tag{B8}$$

$\delta\beta_A^{O3}(r)$ is proportional to the $\left[\frac{1}{B(r)} + 1\right]$ factor and ozone absorption uncertainty, meaning that $\delta\beta_A^{O3}(r)$ is smaller at
longer wavelengths due to larger aerosol scattering ratio and smaller ozone absorption. When $\Delta r$ is specified at 22.5
m, $\delta\beta_A^{O3}(r)$ is less than 0.3%. We still simulate the vertical accumulation of $\delta\beta_A^{O3}(r)$ using Eq. (B4). As noted earlier,
the systematic errors of the DIAL ozone measurement tend to accumulate while the random errors tend to cancel out.
The dominant error source for lidar measurements at the far range is typically the lidar signal detection noise, a type
of random error. Therefore, for purposes of estimation, we assume a 5% constant DIAL retrieval uncertainty primarily
covering the uncertainties due to ozone absorption cross section, non-ozone gas interference, and signal saturation
effect (Leblanc et al., 2018; Wang et al., 2017). As shown in Figure B1, the simulated aerosol retrieval uncertainty
due to ozone is relatively minor and is less than 5% at most altitudes.
In summary, the uncertainties in aerosol backscatter retrieval for the ozone lidar are controlled by $\Delta S$ at near
ranges (i. e., in the PBL) where the air is most turbid and are determined by both the lidar signal detection error and
inaccurate boundary value assumption at far ranges (higher than 7 km) where the air is typically clear. In the middle
range of the lidar measurement (PBL top – 7 km), the air density calculation error may become a significant error
source for aerosol retrieval and may have a comparable influence on the aerosol retrieval as the lidar signal
measurement error. Relative to the above four uncertainty sources, ozone DIAL retrieval error is relatively
unimportant, especially in the lower altitudes where lidar SNR is large enough. All the uncertainty terms are affected
by the aerosol-to-molecular backscatter ratio, $B(r)$, which represents the relative importance of the aerosol component
in both extinction and backscatter processes. Based on the above uncertainty budget analysis, we conclude that the
RO$_3$QET lidar is capable of measuring aerosol profile reliably below 6 km with the current laser output power.

**Acknowledgements**
The authors thank the Tropospheric Composition Program of the National Aeronautics and Space Administration
(NASA)'s Science Mission Directorate for supporting the TOLNet program. A portion of the research was carried out
at the Jet Propulsion Laboratory, California Institute of Technology, under a contract with NASA (80NM0018D0004).
The views, opinions, and findings contained in this report are those of the authors and should not be construed as an
official NASA, National Oceanic and Atmospheric Administration, or U.S. Government position, policy, or decision.

**Data availability**
The ozone lidar data are available at the TOLNet website, https://www-air.larc.nasa.gov/missions/TOLNet/. The
HSRL data used in this study can be obtained at the University of Wisconsin lidar website, http://hsrl.ssec.wisc.edu/.

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

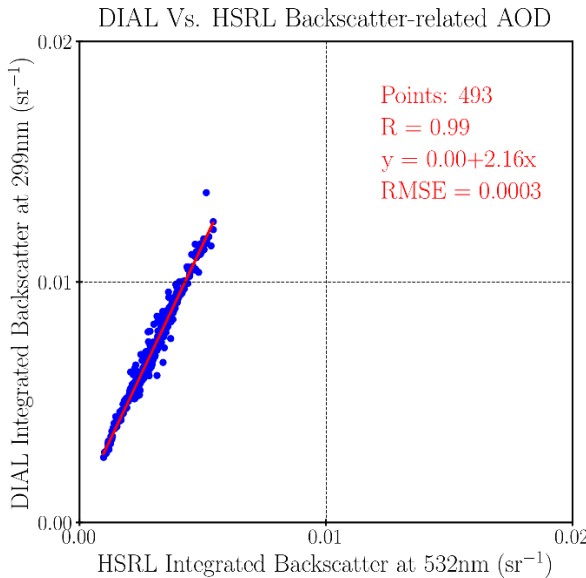


Figure 1. Regression of ozone DIAL and HSRL derived integrated aerosol backscatter between 0.4 and 6 km asl using
the best least-squares fit, resulting in a backscatter color ratio of 1.34 for 299–532-nm for four cases in 2013. All the
data was taken at Huntsville, AL, USA, during summer 2013.

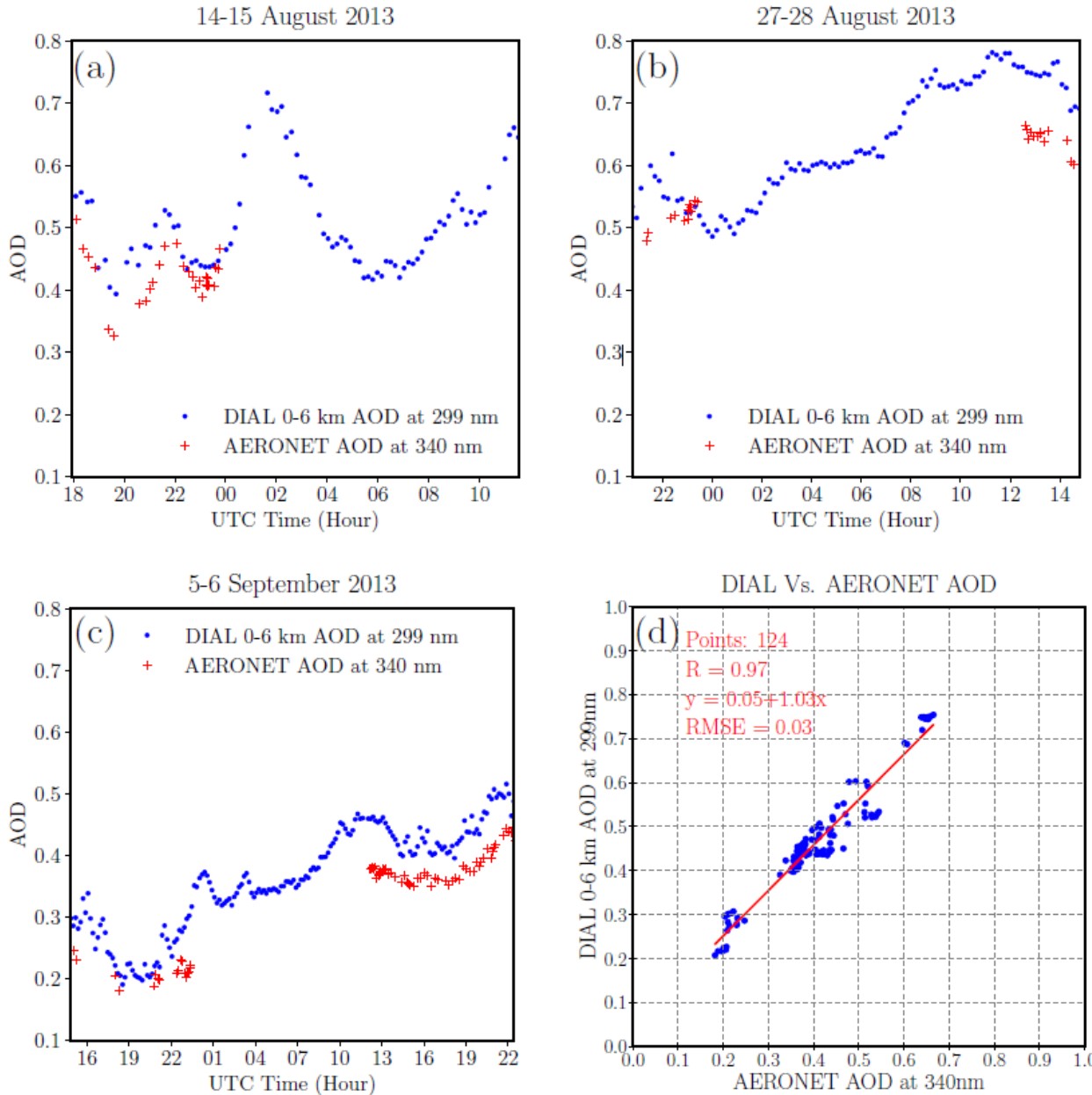


Figure 2. RO$_3$QET DIAL derived AOD between 0 and 6 km at 299 nm using *S*=60 sr compared to collocated
AERONET AOD at 340 nm for (a) 14–15 August, (b) 27–28 August, and (c) 5–6 September 2013. (d) Regression of
the paired data after the DIAL AOD is interpolated to the times of AERONET AOD measurements.

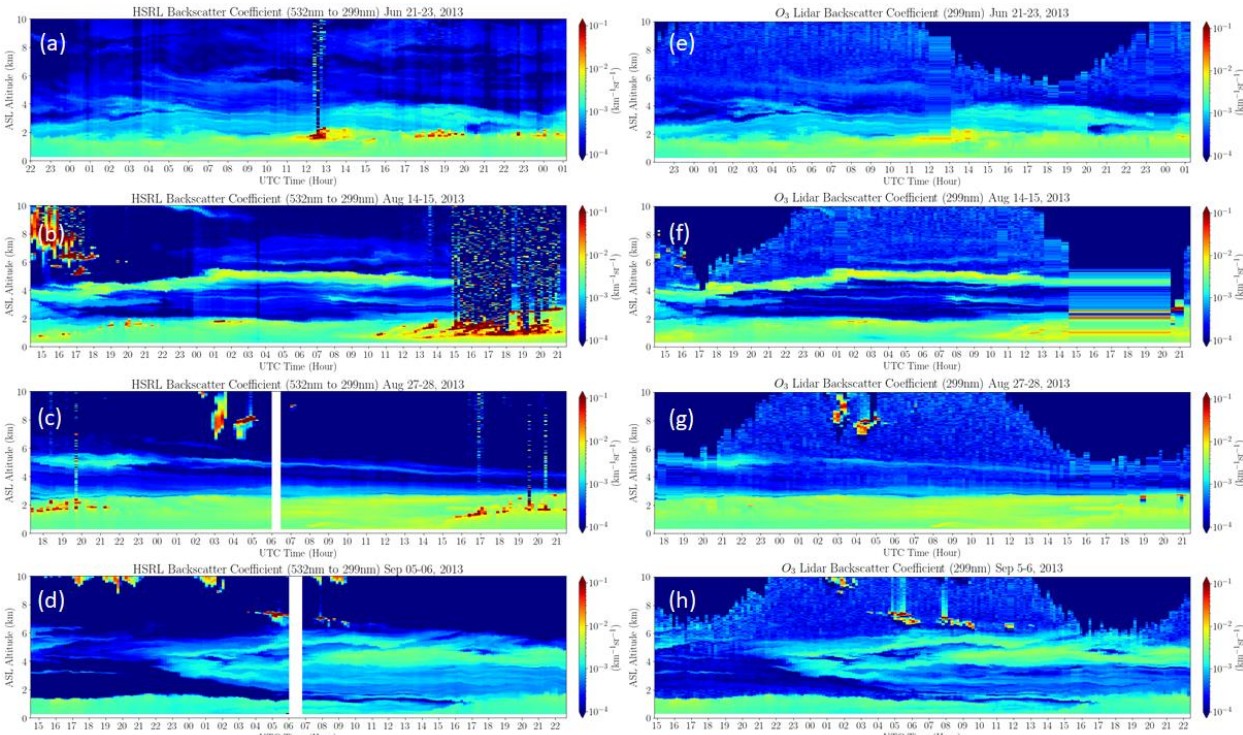


Figure 3. HSRL-converted aerosol backscatter coefficients (a, b, c, d) compared to RO$_3$QET lidar-derived aerosol
backscatter coefficients at 299 nm (e, f, g, h), with 10-min temporal resolution and 30-m vertical resolution. The data
was taken from 21–23 June (a, e), 14–15 August (b, f), 27–28 August (c, g), and 5–6 September (d, h) 2013. The
HSRL-converted aerosol backscatter coefficients are scaled from the original retrievals at 532 nm to 299 nm using
Eq. (1) and å$_\beta$=1.34.

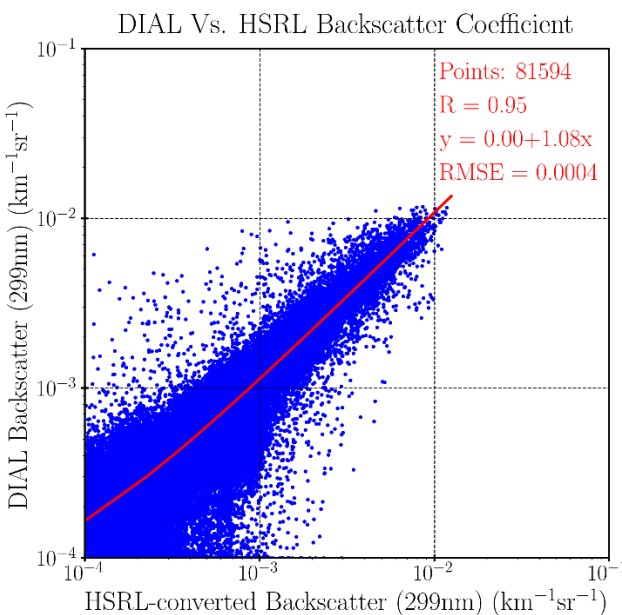


Figure 4. Regression of ozone lidar measured and HSRL-converted aerosol backscatter coefficients (interpolated to
299 nm with å$_\beta$=1.34) with 30-m vertical resolution and 10-min temporal resolution. The regression line is a little
curved in the logarithmic scale because the intercept is not exactly zero.

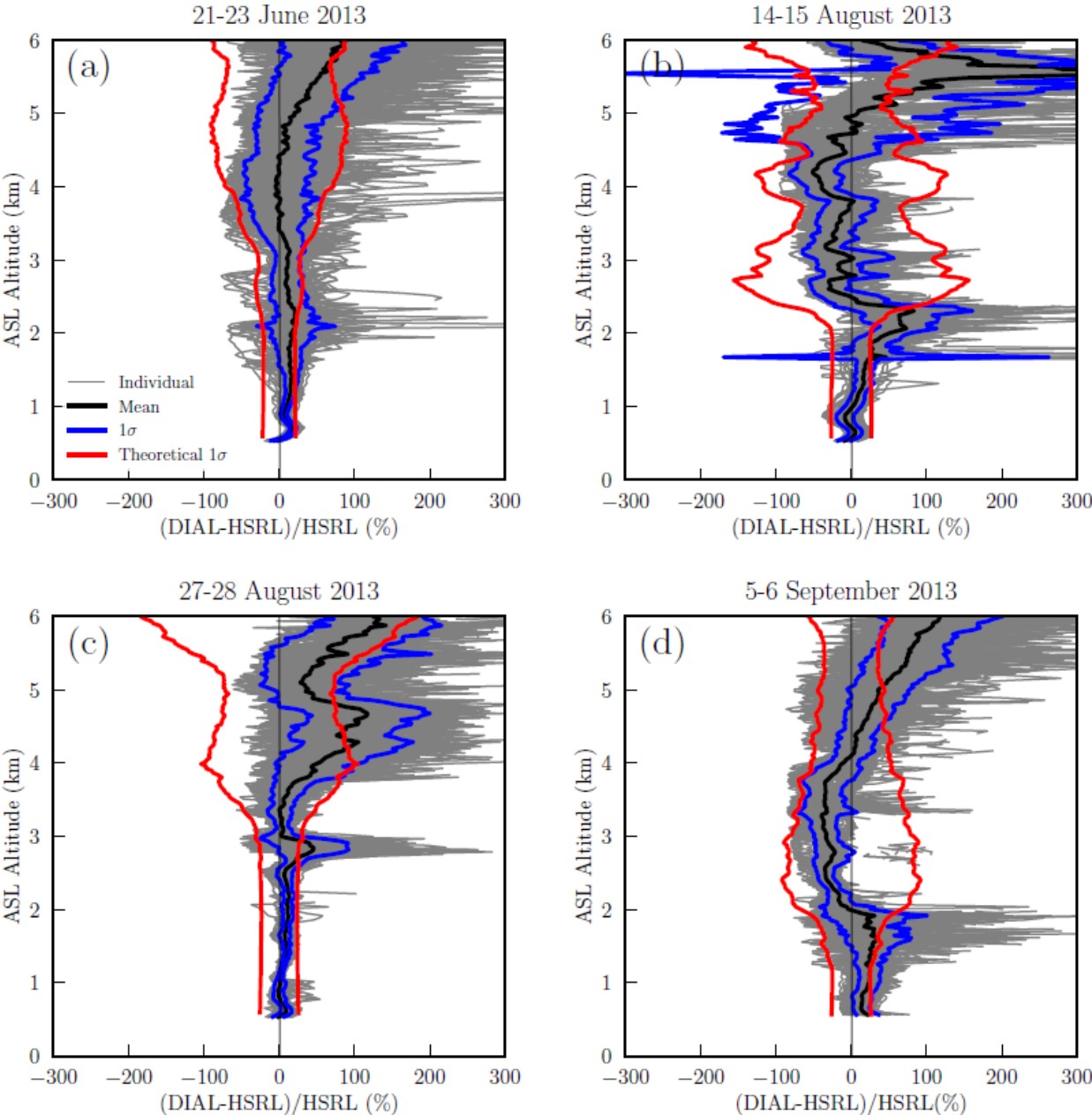

Figure 5. Relative differences between RO₃QET and HSRL-converted aerosol backscatter measurements, (RO₃QET-HSRL)/HSRL, made from (a) 21–23 June, (b) 14–15 August, (c) 27–28 August, and (d) 5–6 September 2013. The gray and black lines represent the differences for the 10-min individual aerosol backscatter profiles and their mean, respectively. The blue lines represent the actual 1 σ of the differences compared to the theoretical 1 σ (red lines) of the RO₃QET lidar aerosol measurement.


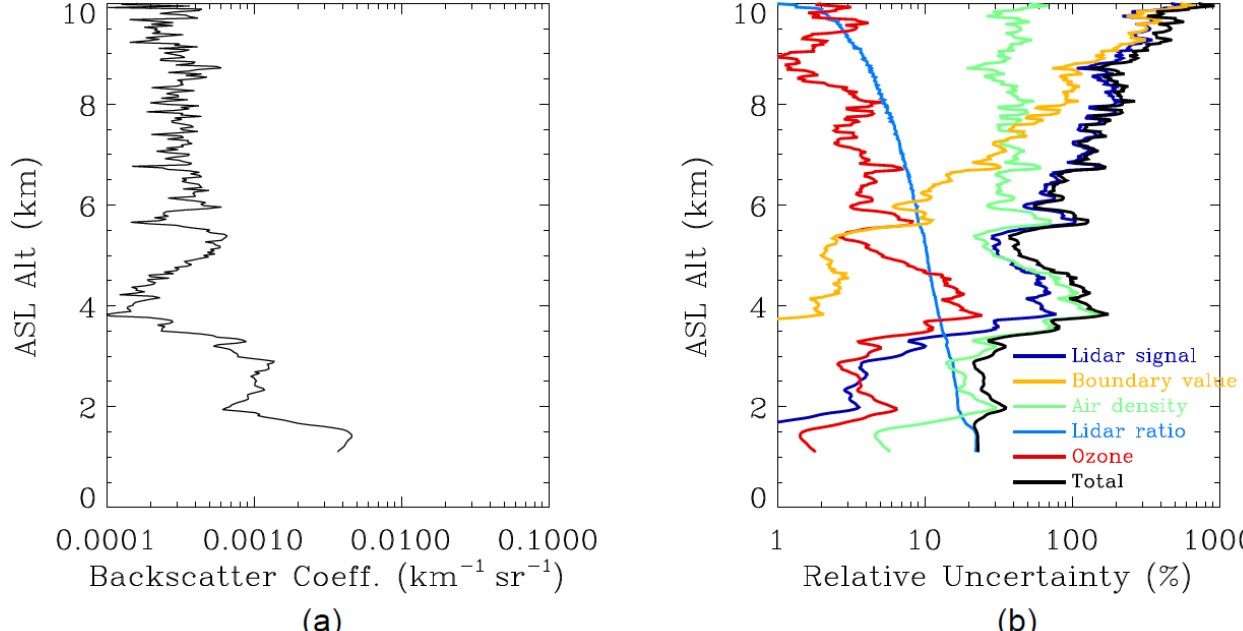

(a)  (b)


Figure B1. An example of (a) aerosol backscatter profile retrieved from 10-min ozone lidar data at about 8:30 UTC
on 22 June 2013 and (b) the retrieval error budget for different uncertainty sources. The lidar data was from the
Channel-3 receiving system, which covered most of the measurement altitude range, and was arbitrarily chosen for a
cloud-free scenario.