# Peer review of "EVALUATION OF UV AEROSOL RETRIEVALS FROM AN OZONE LIDAR"

_Atmospheric Measurement Techniques, 2020_

## Referee Comment (RC1) · Anonymous Referee #1 · 11 May 2020

In the paper titled "EVALUATION OF UV AEROSOL 1 RETRIEVALS FROM AN OZONE LIDAR", the authors described a new approach for retrieving aerosol properties using an ozone lidar (DIAL). The use of an ozone lidar for aerosol retrievals is rather interesting yet I have some issues with the paper, listed below, that I hope the authors can address.

Several of the parameters, including the lidar ratio and aerosol backscatter color ratio, are a strong function of aerosol type. A lidar ratio of 0.6 is assumed with a 20% uncertainty. The aerosol backscatter color ratio is assumed to be 1.34 with an uncertainty of ±0.11. Note that for different aerosol types, both parameters could change significantly (beyond their mentioned uncertainties). It is unsure if aerosol type could be derived from the proposed method. Without a valid method for retrieving aerosol types, generalized applications of the proposed method may be problematic. The authors should at least clearly illustrate the limitations.

AERONET data are also available from the Huntsville AERONET site. I wonder if the authors could inter-compare AERONET AODs with HSRL/ RO3QET lidar derived AODs. At least the authors should compare HSRL and AERONET AODs.

The retrieved aerosol profiles are used to further refine ozone retrievals. I was wondering if the refined ozone retrievals can be further used for refining aerosol retrievals.

Other comments

Line 121, "as you go towards the" - who is "you"?

Line 141, "10-min temporal average and 30-m spatial average for both HSRL". - Should be "30-m vertical average"?

Lines 151-152, "Therefore, data contaminated by clouds is filtered out. " - What are the cloud screening steps? Those steps need to be included.

Lines 170-171, "The slope of the regression (2.16) results in the best" - what is "(2.16)" referring to?

Line 278, equation A2, need a reference for this equation.

Line 306, equation B1, need a reference for this equation.

Equations B3 and B4. Define $\Delta\beta^{sig}_A(r)$ and $\delta\beta^{sig}_A(r)$.

---

## Referee Comment (RC2) · Anonymous Referee #2 · 7 Jun 2020

**1   General Remarks**

The authors give a comprehensive description of their approach for retrieving aerosol backscatter profiles from the return signals of an atmospheric lidar operating in the UV near 290 nm. Their aerosol results in the UV are compared with HSRL lidar measurements of aerosol at 532 nm. Generally good agreement is found. Uncertainties of the retrieved aerosol properties in the UV are also estimated. They usually exceed 50% over a wide range of altitudes.

I agree with reviewer 1 that comparison to Aeronet optical depth data would be a good addition to the paper. I also agree with reviewer 1 that a few more cautionary remarks on the variation of extinction to backscatter ratio and aerosol wavelength dependences

between aerosol types should be added. However, in many cases the stated large uncertainties probably cover a good fraction of these changes between aerosol types.

Overall I think this is a solid paper, well suited to the scope of AMT. I recommend publication with only a few minor revisions.

**2  Suggestions**

line 33: "weighing" or "weighting"?

lines 35/36: I suggest to add the Browell et al. 1985 reference here as well. Ed Browell really pioneered operational airborne UV-lidar measurements of tropospheric ozone in the 1980s.

lines 32 to 42: Here, and in several other places of the paper (e.g. lines 256 to 262), I suggest to add more cautionary sentences on the general problem of aerosol interference on DIAL ozone measurements (Browell et al. 1985, but also Steinbrecht and Carswell, JGR, 1994). Especially the differential backscatter term can cause large problems for narrow aerosol layers (errors exceeding 10s of percents). Investigations of aerosol effects on ozone, of the order of a few percent, are very desirable, but substantial caution is required.

lines 48/49: there is ozone absorption at 532 nm, which is not necessarily negligible. Add statement.

lines 55 to 62: Maybe the authors should move this to the beginning of the paragraph, and even extend it? Important lidar facts are: Because of the strong wavelength dependence of molecular Rayleigh scattering ($\lambda^{-4}$), and the weaker wavelength dependence of aerosol scattering $\approx \lambda^{-1.5}$, aerosol is measured best by lidars at 532 and 1064 nm (NdYAG) or 694 nm (Ruby). Nevertheless, the authors' UV lidar also measures aerosol, and aerosol interference on the ozone measurement needs to be

looked at. Fortunately, because of the large increase of ozone extinction from 320 nm to 250 nm (2 orders of magnitude), aerosol interference at your wavelengths (around 290 nm) is a factor of 5 to 20 smaller than, e.g., for a stratospheric ozone DIAL (around 310 nm) for the same amount of aerosol.

line 154: "owning" -> "owing"?

line 186: Why is the extinction wavelength exponent (1.49) different from the backscatter wavelength exponent (1.34). Is that because the constant lidar ratio ($S = 55$) is only used for the UV lidar, whereas the HSRL actually measures extinction? The authors might want to clarify that. I also wonder how meaningful this entire extinction comparison then is, because the UV lidar really only measures backscatter, and extinction is largely assumed.

lines 225 to 236: In Fig. 2, the UV lidar measured backscatter above 6 km during daytime clearly is too high (lighter blue colors). The authors state that they are not looking at these altitudes. However, I am wondering if the systematic high bias above 5 km in Fig. 4, has something to with the daytime high bias above 6 km in Fig. 2? Have the authors considered that? A few additional sentences might be good.

Figure 3: Maybe a logarithmic plot (both axes) would be better here? A lot of the data points are close to the lower left corner (0,0).

―――――――――――――――――

---

## Author Comment (AC1) · 24 Jul 2020

**Author Responses to Referee 1**

We would like to thank both reviewers for their significant time spent reading the manuscript and for their constructive comments for improving it. We have made very careful revisions to address all these comments. In the following paragraphs, the **bolded** words represent the reviewers' original comments and the unbolded text represents our answers.

**Referee 1:**

**In the paper titled "EVALUATION OF UV AEROSOL RETRIEVALS FROM AN OZONE LIDAR", the authors described a new approach for retrieving aerosol properties using an ozonelidar (DIAL). The use of an ozone lidar for aerosol retrievals is rather interesting yet I have some issues with the paper, listed below, that I hope the authors can address.**

**Several of the parameters, including the lidar ratio and aerosol backscatter color ratio, are a strong function of aerosol type. A lidar ratio of 60 is assumed with a 20% uncertainty. The aerosol backscatter color ratio is assumed to be 1.34 with an uncertainty of ±0.11. Note that for different aerosol types, both parameters could change significantly (beyond their mentioned uncertainties). It is unsure if aerosol type could be derived from the proposed method. Without a valid method for retrieving aerosol types, generalized applications of the proposed method may be problematic. The authors should at least clearly illustrate the limitations.**

**AERONET data are also available from the Huntsville AERONET site. I wonder if the authors could inter-compare AERONET AODs with HSRL/ RO3QET lidar derived AODs. At least the authors should compare HSRL and AERONET AODs. The retrieved aerosol profiles are used to further refine ozone retrievals. I was wondering if the refined ozone retrievals can be further used for refining aerosol retrievals.**

We have accepted the reviewer's suggestion and added a paragraph on $RO_3QET$ data evaluation using collocated AERONET data at 340 nm (Lines 213-237 in the change-tracked version). The AOD from $RO_3QET$ and AERONET are highly correlated, with r = 0.97. The $RO_3QET$ AOD is on average 15% larger than the AERONET AOD due to the shorter wavelength of the lidar measurement, suggesting that the choice of $S$ = 60 sr is very reasonable. For a rough estimation, the one-sigma standard deviation (9%) of the differences can be considered as the uncertainty for $S$ if the variability of these differences are mostly due to the variation in $S$. (If $S$ has uncertainty > 9%, we expect the one-sigma value of the differences to be larger than 9% since AERONET measures extinction and $RO_3QET$ directly measures backscatter.) Considering that AERONET measures the column average AOD, with longer temporal integration, has its own uncertainty, and covers only 38% of the total observational period, the ±20% uncertainty of $S$ for a higher vertical resolution measurement should be large enough to cover various uncertainty sources. Therefore, the additional AERONET data not only convinces us that the data quality of both instruments is good, but also confirms that the assumed $S$ *a priori* and uncertainty are appropriate.

We agree with the reviewer that the lidar ratio changes with aerosol type and that it is hard to derive the aerosol type from elastic lidar measurements. We have provided caveats and limitations of this work in multiple places in the text. For example, in Lines 111-113, we say: **"**The $S$ *a priori* value assumed for this study represents a mix of urban and smoke aerosols during the lidar observations (Ackermann, 1998; Burton et al., 2012; Cattrall et al., 2005; Groß et al., 2013; Müller et al., 2007). The *a priori* is application dependent." Further, in Lines 326-329 in the Conclusions section, we say: "These exponents represent a summertime average for a mixture of urban pollution and fire smoke. Speciation of aerosol types was not done in this work, although we recognize that $S$ and Ångström exponent vary with the aerosol phase function and size distribution.**"**

1. **Other comments Line 121, "as you go towards the" -who is "you"?**

To address this issue, we have changed "decreases as you go towards the ground from the far range" to "decreases towards the ground from the far range".

**2. Line 141, "10-min temporal average and 30-m spatial average for both HSRL". -Should be "30-m vertical average"?**

We have replaced "spatial" with "vertical" as per suggestion.

**3. Lines 151-152, "Therefore, data contaminated by clouds is filtered out. "-What are the cloud screening steps? Those steps need to be included.**

The cloud screening process is described at Line 89-95 in Section 2.1. We have added additional description to clarify our cloud treatment: "The cloud base height is determined by the following empirical method. Derivatives of the logarithm of the off-line analog signal are calculated for a lidar signal profile and the first range bin at which the derivative is greater than a certain threshold is considered to be the cloud base height. The threshold is chosen empirically based on the lidar SNR and the vertical resolution. Lidar data with cloud base lower than 2 km was discarded."

**4. Lines 170-171, "The slope of the regression (2.16) results in the best" -what is "(2.16)" referring to?**

To clarify, we change this sentence to "the slope of the regression, equal to 2.16, results in …".

**5. Line 278, equation A2, need a reference for this equation.**

We have added (Uchino et al., 1980) as the reference for Equation (A2).

**6. Line 306, equation B1, need a reference for this equation. Equations B3 and B4. Define ΔβsigA(r) and δβsigA(r).**

We have added (Taylor, 1997) as the reference for Equation (B1) a Line 374.

At Line 374, we made a change to reflect the definition of $\Delta\beta^{sig}_A(r)$ as: "…we obtain the uncertainty of the aerosol backscatter owing to lidar signal measurement error, $\Delta\beta^{sig}_A(r)$, relative to the total backscatter as…".

The definition of $\delta\beta^{sig}_A(r)$ is already stated at Line 387 ahead of Equation (B3), so there is no change for that.

---

## Author Comment (AC2) · 24 Jul 2020

**Author Responses to Referee 2**

We would like to thank both reviewers for their significant time spent reading the manuscript and for their constructive comments for improving it. We have made very careful revisions to address all these comments. In the following paragraphs, the **bolded** words represent the reviewers' original comments and the unbolded text represents our answers.

**Referee 2:**

**1 General Remarks**

**The authors give a comprehensive description of their approach for retrieving aerosol backscatter profiles from the return signals of an atmospheric lidar operating in the UV near 290 nm. Their aerosol results in the UV are compared with HSRL lidar measurements of aerosol at 532 nm. Generally good agreement is found. Uncertainties of the retrieved aerosol properties in the UV are also estimated. They usually exceed 50% over a wide range of altitudes.**

**I agree with reviewer 1 that comparison to Aeronet optical depth data would be a good addition to the paper. I also agree with reviewer 1 that a few more cautionary remarks on the variation of extinction to backscatter ratio and aerosol wavelength dependences between aerosol types should be added. However, in many cases the stated large uncertainties probably cover a good fraction of these changes between aerosol types.**

**Overall I think this is a solid paper, well suited to the scope of AMT. I recommend publication with only a few minor revisions.**

We thank the reviewer for the positive comments and good suggestions. We have added a paragraph on RO$_3$QET data evaluation using collocated AERONET data at 340 nm (Lines 213-237) to address both reviewers' suggestion. The AERONET data provide a very nice evaluation of the ozone lidar data and the assumed lidar ratio. (Please refer to the answers to Reviewer 1 for more details.) The references for AERONET have been also added.

**2 Suggestions**

**1.  line 33: "weighing" or "weighting"?**

We meant "weighing", so we keep it unchanged.

**2.  lines 35/36: I suggest to add the Browell et al. 1985 reference here as well. Ed Browell really pioneered operational airborne UV-lidar measurements of tropospheric ozone in the 1980s.**

We agree and we have added Browell et al. 1985 in the citation list here.

**3.  lines 32 to 42: Here, and in several other places of the paper (e.g. lines 256 to 262), I suggest to add more cautionary sentences on the general problem of aerosol interference on DIAL ozone measurements (Browell et al. 1985, but also Steinbrecht and Carswell, JGR, 1994). Especially the differential backscatter term can cause large problems for narrow aerosol layers (errors exceeding 10s of percents). Investigations of aerosol effects on ozone, of the order of a few percent, are very desirable, but substantial caution is required.**

We have cited Steinbrecht and Carswell (1994) in the 1$^{st}$ paragraph.

**4.  lines 48/49: there is ozone absorption at 532 nm, which is not necessarily negligible. Add statement.**

We have replaced "negligible" with "much smaller than".

5. **lines 55 to 62: Maybe the authors should move this to the beginning of the paragraph, and even extend it? Important lidar facts are: Because of the strong wavelength dependence of molecular Rayleigh scattering ($\lambda^{-4}$), and the weaker wavelength dependence of aerosol scattering $\sim\lambda^{-1.5}$, aerosol is measured best by lidars at 532 and 1064 nm (NdYAG) or 694 nm (Ruby). Nevertheless, the authors' UV lidar also measures aerosol, and aerosol interference on the ozone measurement needs to be looked at. Fortunately, because of the large increase of ozone extinction from 320 nm to 250 nm (2 orders of magnitude), aerosol interference at your wavelengths (around 290 nm) is a factor of 5 to 20 smaller than, e.g., for a stratospheric ozone DIAL (around 310 nm) for the same amount of aerosol.**

We have made change to say: "Lasers used for aerosol lidars are preferred in the visible and infrared bands, typically 532 and 1064 nm for Nd:YAG laser or 694 nm for Ruby laser (Russell et al., 1979),…".

We agree with the reviewer on the aerosol interference in the ozone DIAL retrieval approximately proportional to $\Delta\lambda/(\lambda\Delta\alpha_{O3})$. The aerosol interference in DIAL is pretty complicated and is worth another paper to discuss. Since the major purpose of this article is to discuss aerosol retrieval and its uncertainty, we decide not to add more discussion on the aerosol interference in DIAL retrieval. But, this is certainly an important motivation to do aerosol retrieval. So, in the Introduction, we say "Vertical aerosol profiles are of high interest not only because they are needed for aerosol correction in ozone lidar retrievals (Steinbrecht and Carswell, 1994), …". In the Conclusions, we write "Aerosol correction for ozone lidar retrievals will be described in a subsequent paper."

6. **line 154: "owning" -> "owing"?**

We have made the correction as per the suggestion.

7. **line 186: Why is the extinction wavelength exponent (1.49) different from the backscatter wavelength exponent (1.34). Is that because the constant lidar ratio (S = 55) is only used for the UV lidar, whereas the HSRL actually measures extinction? The authors might want to clarify that. I also wonder how meaningful this entire extinction comparison then is, because the UV lidar really only measures backscatter, and extinction is largely assumed.**

Yes. The extinction wavelength exponent is different from the backscatter exponent because the lidar ratio (*S*) is wavelength dependent. After an extensive literature review, we assumed S=60 sr at 299 nm and the resulting extinction retrievals agree well with the AERONET observations at 340 nm (we have added this description in the AERONET-DIAL comparison). The *S*=55 sr assumption for the HSRL at 532 nm is taken from Reid et al. 2017, who derived this value by comparing HSRL data (532 nm) with collocated AERONET observations at 500 nm. To clarify this, we have added "the calculated Ångström exponent is different from the backscatter-related wavelength exponent because of the wavelength dependence of *S*."

Yes, as the reviewer pointed out, the elastic lidar directly measures backscatter so that the extinction retrieval has larger uncertainty than backscatter retrieval. However, we still believe the extinction retrieval is meaningful because "In practice, aerosol extinction is a more meaningful parameter and more relevant for several applications than backscatter" (Lines 203), such as AOD calculations to compare with satellite data.

8. **lines 225 to 236: In Fig. 2, the UV lidar measured backscatter above 6 km during daytime clearly is too high (lighter blue colors). The authors state that they are not looking at these altitudes. However, I am wondering if the systematic high bias above 5 km in Fig. 4, has something to with the daytime high bias above 6 km in Fig. 2? Have the authors considered that? A few additional sentences might be good.**

We agree that the large positive differences above 5 km can be due to strong solar background during daytime. We have added the following explanation:

"These positive biases can be caused by two reasons. First, the RO$_3$QET derived aerosol extinction above 5 km is obviously larger than that from HSRL during daytime due to the solar background impact, which is especially strong in the summer. The relative differences are even worse in clean (compared to turbid) regiosn during the daytime because of the small number division effect mentioned earlier. It can be seen from Figure 3 that RO$_3$QET nighttime retrievals above 5 km and daytime retrievals below 5 km are relatively good due to either lower solar background or larger lidar signal resulting in better SNR."

9. **Figure 3: Maybe a logarithmic plot (both axes) would be better here? A lot of the data points are close to the lower left corner (0,0).**

We have accepted the reviewer's suggestion and changed the linear scale to log scale for Figure 3 (currently Figure 4). There are about 80,000 points in that figure, with a large number being very small values. So, the log scale is better to show how the points are scattered relative to the regression line.